# Community-based fact-checking reduces the spread of misleading posts on X (formerly Twitter)

Yuwei Chuai [1], Moritz Pilarski [2], Thomas Renault[3], David Restrepo-Amariles[4], Aurore Troussel-Clément[4], Gabriele Lenzini [1] & Nicolas Pröllochs [2] ✉

Community-based fact-checking is a promising approach to correct misleading posts at scale. Yet, causal evidence regarding its effectiveness in reducing the spread of misinformation on social media is missing. Here, we perform a large-scale empirical study to analyze whether community notes reduce the spread of misleading posts on X (formerly Twitter). Using a Difference-in-Differences design and repost time series data for $N$ = 237,180 (community fact-checked) cascades that have been reposted more than 431 million times, we find that exposing users to community notes reduces the subsequent spread of misleading posts by, on average, 61.2%. The effect is pronounced across the board but significantly weaker for posts from influential accounts and political content. Additionally, community notes increase the odds that users delete their misleading posts by 94.3%. Although community notes are broadly effective in reducing the spread of posts once annotated, they often appear too late to intervene in the early (and most viral) stage of the diffusion. As a result, their system-wide effect is more modest, lowering total engagement with misleading posts by 14.9%. Our work provides important insights that can inform future initiatives aimed at increasing the effectiveness of community-based fact-checking approaches on social media.

The spread of misinformation on social media platforms has become a concerning issue of the digital age and raised alarm bells across various domains. Negative repercussions of misinformation have been repeatedly observed, with tangible consequences in critical areas, such as elections[1–6], public health[7–10], and public safety[11–13]. Researchers, governments, and regulation authorities thus urge social media providers, such as X (formerly Twitter) and Facebook, to develop effective countermeasures to counteract the spread of misinformation on their platforms[14–18].

To this end, a widely implemented approach is the use of professional fact-checkers to identify and label misleading posts[19,20]. The rationale is that if users are warned that a message is false, they should be less likely to believe it. While this approach has been shown to be effective in numerous experimental studies[21–31] (for a review, see

ref. [31]), it faces several key challenges, particularly with regard to volume, visibility, and trust. First, in/depth investigation of claims can be time/consuming, often taking many hours or even days[32]. Given the limited number of available fact/checkers, not every claim can be thoroughly examined. Consequently, fact/checkers are compelled to prioritize content that is overtly false or intentionally misleading over material that is more intricate or nuanced[33]. Second, professional fact/checks frequently have a restricted audience. Except for some collaborations with social media platforms on particular topics, fact/checking organizations primarily disseminate their findings through their own websites. A study conducted in 2017 revealed that over half of all U.S. adults had never visited any fact/checking website[34], demonstrating the limited reach of professional fact/checks. Third, a major drawback is that professional fact/checkers face a lack of trust

[1]University of Luxembourg, Luxembourg, Luxembourg. [2]JLU Giessen, Giessen, Germany. [3]Université Paris-Saclay, Paris, France. [4]HEC Paris, Paris, France. ✉e-mail: nicolas.proellochs@wi.jlug.de

from large portions of society[35,36]. For example, according to surveys, the majority of Republican partisans (70% vs. 29% of Democrats) and half of all U.S. adults believe that fact/checkers exhibit bias[36].

As a remedy, research has proposed to employ nonexpert fact-checkers in the crowd to fact-check social media content[33,37–43]. This approach is based on the idea that the biases and errors of a single user can be mitigated by tapping into the collective intelligence of diverse individuals, also known as the "wisdom of the crowds"[43–45]. The wisdom of crowds has been repeatedly observed in a wide range of settings, including online platforms, such as Wikipedia and Stack Overflow, where the crowd ensures relatively trustworthy and high-quality accumulation of knowledge[46]. Experimental studies on crowd-based fact-checking found that the assessments of even relatively small crowds are comparable to those of experts[33,38,39,41–43,47]. Furthermore, crowd-based fact-checking enables the examination of significantly larger volumes and varieties of posts as compared to expert-based assessments[48–51]. Altogether, community-based fact-checking may have the potential to address many of the drawbacks of expert-based fact-checking on social media.

Building upon these encouraging findings, the social media platform X (formerly Twitter) has recently introduced "community notes" (formerly "birdwatch"), a community-based fact-checking system that allows users to assess the accuracy of posts[48,52]. "Community notes" is the first large-scale attempt to implement community-based fact-checking on a major social media platform. It enables enrolled users to flag posts that they believe are misleading and add concise fact-checking explanations, which are then rated by other users for their helpfulness[53]. Notes rated by the community to be helpful then appear directly on the fact-checked misleading post to inform other users. However, empirical evidence on whether displaying community notes reduces the spread of misleading posts on social media is missing so far.

In prior research, the efficacy of fact-checking has mainly been studied in one of two forms: (i) lab/survey experiments evaluating whether flagging misinformation influences misinformation discernment and sharing intentions[22–24,26–29,54], and (ii) field studies analyzing the spread of misleading vs. non-misleading posts[49,55–58]. The former group of studies, i.e., lab/survey experiments, have shown that misinformation flags are effective in increasing the participants' ability to discern between true and false content[24,26,28,54] as well as decreasing their sharing intentions for misinformation[22,23,29,54]. The latter of studies, i.e., field studies, were largely limited to correlational works analyzing the virality of misleading vs. non-misleading posts on social media[49,55–61]. As an example, Vosoughi et al.[56] found that expert fact-checked misinformation is linked to more viral resharing cascades. Still, these works have not quantified the real-world intervention effect of fact-checking, mainly due to data constraints, such as missing time series data on reposts and interventions. Here, we add by analyzing the efficacy of community notes in reducing the spread of misleading posts on the social media platform X (formerly Twitter).

In this work, we perform a large-scale quasi-experimental study to analyze whether community fact-checks reduce the spread of misleading posts on the social media platform X (formerly Twitter). For this, we examine $N = 237,180$ (community fact-checked) cascades that were created within a period of over 20 months from the roll-out of "community notes" on October 6, 2022, to June 11, 2024, and have been reposted more than 431 million times on X (formerly Twitter). To evaluate the efficacy of community notes in reducing the spread of misleading posts on X (formerly Twitter), we collect time series data of repost counts for the community fact-checked posts over 36 h since their creation. We then employ a difference-in-differences (DiD) design and negative binomial regression models to estimate the reduction in reposts for misleading posts after the display of community notes. Our results show that the display of community notes reduces the shares of misleading posts by 61.2% on X (formerly Twitter). As a secondary

analysis, we examine the extent to which the "community notes" feature in its current implementation reduces the spread of misleading posts on X (formerly Twitter) in terms of cumulative repost count. The underlying rationale is that the display timing of the notes might be critical. Even if users are less inclined to share tweets once they are equipped with a community note, the fact-check might arrive too late to intervene in the early and most viral stage of the diffusion, thereby potentially diminishing its effect on overall engagement with misinformation on social media. Consistent with this reasoning, we find that the system-wide effect of community notes is more modest, lowering total engagement with misleading posts by 14.9%.

## Results

### Difference-in-differences estimation

We employed a difference-in-differences (DiD) design and negative binomial regression models to estimate the average treatment effect on the treated (ATT), i.e., the extra repost reduction in the treatment group after the display of community notes compared to the baseline before the display and relative to the control group (see "Methods"). To mitigate potential confounding factors and balance treatment and control groups, we constructed a control group from the source posts without displayed notes through one-to-one matching. The one-to-one matching was conducted based on the variables from user profiles (e.g., followers and followees) and post features (e.g., sentiments and topics). As a result, the source posts in the control group have no statistically significant difference with the source posts in the treatment group with respect to the user and post characteristics (see details in Supplementary Note 4). Subsequently, we assigned the virtual time of note display to the source posts in the control group according to the corresponding posts in the treatment and recenter the repost timelines of the source posts around the display of community notes in the treatment and control groups.

We started our analysis with a two-period DiD model. This allowed us to estimate the difference in outcomes (i.e., repost counts) between the treatment and control group before and after the treatment (i.e., displaying community notes) was applied. To this end, we considered the period between 1 and 12 h from the note display as the after-display period and the period between 1 and 4 h before the note display as the before display period (see "Methods"). The ATT for the two-period DiD model is visualized in purple color in Fig. 1 (see Supplementary Note 5 for full estimation results). The estimate for the ATT was −0.612 (99% CI: [−0.617, −0.608]; $z = −211.71$, $p < 0.001$), which indicates that displaying community notes reduced the subsequent number of reposts by, on average, 61.2%. This suggests a pronounced treatment effects of community notes in reducing the spread of misleading posts on X (formerly Twitter).

Subsequently, we estimated a multi-period DiD to enable a temporal analysis of the treatment effect. Here, we again considered a 4-h window before the note display as the before-display period and then examined the hourly multi-period ATTs from 1 to 12 h after the note display (see "Methods"). Equivalence testing[62–64] confirms that the ATT estimates in the hours before note display were small enough to be considered negligible, indicating the absence of meaningful pre-trends and supporting the validity of the DiD framework (see Supplementary Note 5.2). Figure 1 visualizes the hourly multi-period ATTs over time (see Supplementary Note 5 for full estimation results). We observed statistically significant and pronounced ATTs after the display of community notes. During the first 1 h after the note display, the ATT was estimated as −0.363 (99% CI: [−0.380, −0.346]; $z = −43.82$, $p < 0.001$). This implies that community notes reduced the number of reposts by 36.3% within the first hour. Furthermore, the efficacy of community notes increased over time following the note display. The ATTs were −0.530 (99% CI: [−0.542, −0.517]; $z = −72.23$, $p < 0.001$) at the second hour, −0.613 (99% CI: [−0.623, −0.602]; $z = −89.27$, $p < 0.001$) at the fourth hour, and −0.644 (99% CI: [−0.654, −0.634]; $z = −94.71$,

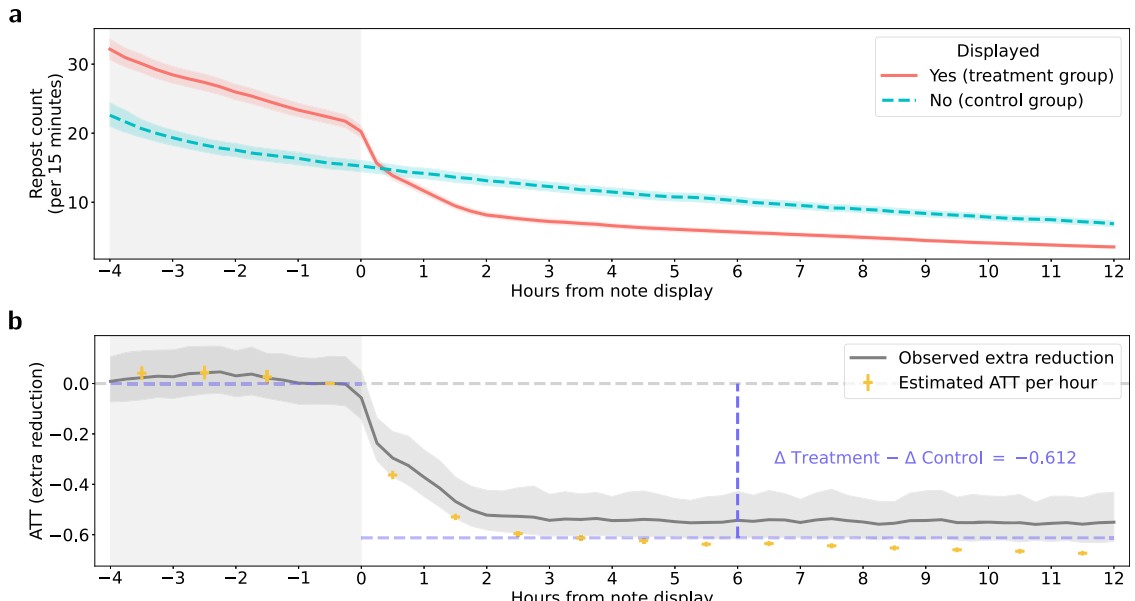

**Fig. 1 | Community notes reduce the spread of misleading posts on X (formerly Twitter). a** Time series of mean repost counts within 15 min intervals in the treatment group (red) and the control group (blue) from 4 h before the display of community notes to 12 h after the display of community notes. The error bands represent 99% confidence intervals (CIs) derived using the bootstrap method for 1000 resamples. **b** Two-period (purple) and multi-period (yellow) ATTs estimated using a difference-in-differences (DiD) design and negative binomial regression models. For the two-period DiD model, the ATT is calculated as Δ treatment − Δ control = −0.612. For the multi-period DiD model, the yellow circles and error bars show the estimated hourly multi-period ATTs (with 99% CIs). The grey band (with 99% CIs) visualizes the observed extra reduction of the ratio of reposts in the treatment group relative to reposts in the control group and compared to the ratio of reposts before the display of community notes. The ATT estimations are based on 654,400 repost time series observations for $N = 40,900$ posts. The 99% CIs were derived using the bootstrap method for 1000 resamples. Post-level random effects are included. Full estimation results are in Tables S5–S8.

$p < 0.001$) at the eighth hour. At the twelfth hour after the note display, the hourly ATT was −0.674 (99% CI: [−0.683, −0.664]; $z = − 100.01$, $p < 0.001$). In sum, our temporal analysis indicates that community notes reduced the number of reposts from 36.3 to 67.4% within the first 12 h after they were displayed to users.

We conducted a comprehensive set of checks to ensure the robustness of our DiD estimates: (i) we performed a placebo test (Supplementary Note 6) by replacing the treatment group with a placebo group. The results showed no additional reduction in reposts within the placebo group, which supports that the estimated ATTs in the treatment group can be attributed to the display of community notes. (ii) To assess the impact of potential violations of parallel trends, we applied HonestDiD, a robust inference approach that quantifies uncertainty in DiD estimates[63,65]. Our estimates remained stable even when allowing post-display deviations to be twice as large as the largest pre-display deviation (Supplementary Note 5.2.2). This demonstrates that our findings hold even under relaxed parallel trends assumptions. (iii) We implemented a battery of additional checks, including alternative model specifications, estimators, and timeframes (Supplementary Note 10), all of which consistently supported the robustness of our findings.

Furthermore, we complemented our main analysis with a wide range of sensitivity analyses (Fig. 2). First, we studied how the efficacy of community notes varied depending on the post age at note display (Fig. 2a). Here, we found that the efficacy of community notes was larger if they were displayed earlier. Second, we examined changes in the efficacy of community notes since the roll-out of the feature in October 2022. As shown in Fig. 2b, we observed a trend of increasing efficacy of community notes in the months following its launch. This may be partly attributed to improvements within the community notes program, such as faster note generation (see Supplementary Note 12). Third, we examined the role of helpfulness ratings (Fig. 2c) and found that community notes with higher helpfulness scores (i.e, higher agreement among raters) had a larger efficacy.

Fourth, we analyzed whether the treatment effect of community notes was moderated by author characteristics (e.g., the number of followers, verified status) and post characteristics (e.g., sentiment, topics; see Fig. 2d). We found that the efficacy of community notes was smaller for source posts from verified users and high-follower accounts. This suggests that the effectiveness of community notes might be slightly discounted for source posts from accounts with high social influence. Regarding post characteristics, we observed that community notes had a smaller efficacy for posts that had media elements and if they were they were attached to posts covering health-related and political topics. Notably, despite these statistically significant variations (each $p < 0.001$), the treatment effect remained statistically significant and pronounced across the board. This points towards a broad effectiveness of community-based fact-checking on social media. The full results of our sensitivity analyses can be found in Supplementary Note 7.

### Effect on the cumulative repost count
The results of our DiD analysis imply that displaying community notes can reduce the spread of misleading posts on X (formerly Twitter) by, on average, 61.2% compared to the control group and relative to the before-display period. However, compared to the rapid dissemination of posts on X (formerly Twitter), fact-checking via community notes was relatively slow (see Supplementary Note 1). While 75.7% of all notes deemed helpful were displayed to general users within 36 h after their creation, the time lag between post creation and note display (i.e., the response time) averaged at 62.9 h (median of 18.1 h). In contrast, posts on X (formerly Twitter) spread considerably faster: the average half-life (i.e., the post age at which the cumulative ratio of reposts reaches to 50% of 36-h reposts) amounted to merely 6.25 h for posts with displayed notes. As a consequence, community notes, on average, might have come too late to reduce the spread of misleading posts at the most viral stages of diffusion. To further scrutinize the system-level effect of community notes, we analyzed the share of the cumulative

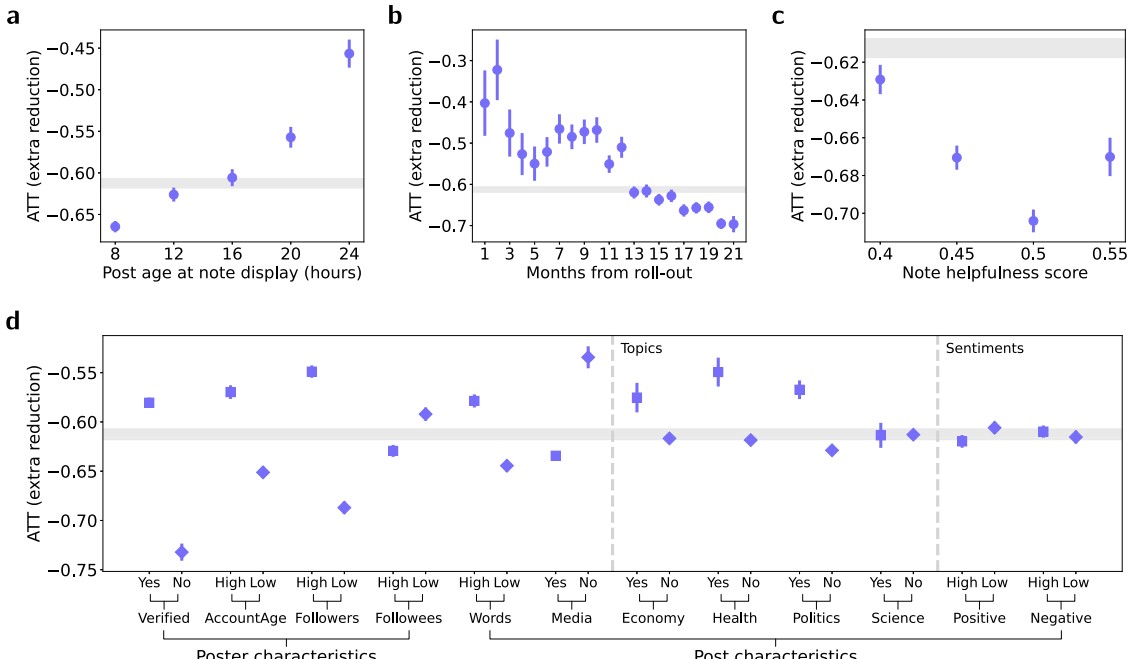

**Fig. 2 | Sensitivity analysis. a** The estimated ATTs (circles) across different response times from post creation to note display (grouped within 4-h windows). Full regression results and sample sizes are reported in Tables S17 and S18. **b** The estimated ATTs (circles) of community notes across the months following the roll-out of the "community notes" program in October 2022. Full regression results and sample sizes are reported in Tables S19 and S20. **c** The estimated ATTs (circles) of community notes depending on helpfulness scores. Full regression results and sample sizes are reported in Tables S25 and S26. **d** The estimated ATTs (circles) within subgroups separated by poster and post characteristics. Full regression results and sample sizes are reported in Tables S27–S32. In all plots, the error bars represent 99% CIs derived using the bootstrap method for 1000 resamples; the grey bands visualize the ATT (with 99% CIs) estimated via the two-period DiD model from our main analysis.

repost count for misleading posts actually prevented by the community notes on X (formerly Twitter).

To this end, we used our DiD models to predict the number of reposts (at 15-min intervals) that the treated posts would have received in the absence of displayed notes. Subsequently, we compared the ratio of the predicted cumulative repost count (over 36 h) in the absence of displayed notes to the actual cumulative number of reposts (over 36 h). Figure 3a shows the actual vs. predicted ratio across different response times from post creation to note display. As expected, we found that the ratio of reposts prevented due to community notes decreased over the response time, i.e., was lower for late vs. early notes. The average reduction in the actual vs. predicted ratio is −0.149, which means that community notes reduced the overall number of reposts on X (formerly Twitter) by merely 14.9% (median of 10.1%). Similarly, in absolute numbers, we only observed a reduction of, on average, 430 reposts between the actual cumulative repost count (mean of 1792) compared to the predicted cumulative repost count (mean of 2222; KS = 0.039, $p < 0.001$; see Fig. 3b, c).

Additionally, we simulated the potential reduction in the cumulative repost count if all community notes would have been displayed at specific times from 2 to 36 h after post creation. As shown in Fig. 3d, if community notes would have been displayed on misleading posts at the second hour since post creation, they would have reduced the overall repost count by 52.3% ($p < 0.001$). The reduction gradually decreased with increasing post ages at note display and became statistically insignificant if more than 24 h had passed since the creation of the posts.

## Analysis of reposting mechanisms

To shed further light on the causal mechanisms underlying the reduction in resharing of community-noted posts, we used the X (formerly Twitter) API to collect additional information on a random

subset of users who reshared posts with community notes (see Supplementary Note 9). This reposter dataset (across $N = 3163$ posts) allowed us to analyze four potential mechanisms through which community notes may influence resharing behavior among reposters: (i) prior interactions with the authors of the fact-checked post (e.g., reduced sharing by dedicated followers), (ii) the verification status on X (formerly Twitter) (e.g., reduced sharing by verified users), (iii) the political orientation of both posters and reposters (e.g., reduced sharing for politically concordant posts), and (iv) prior exposure to misinformation on X (formerly Twitter) (e.g., reduced sharing by frequent consumers of misinformation).

The estimated ATTs across the four reposter characteristics are visualized in Fig. 4a (see Supplementary Note 9 for full estimation results). Our results imply that a key mechanism by which community notes reduced engagement was by curbing resharing by users with no prior interaction (i.e., disconnected users) with the authors of the misleading posts. Specifically, the efficacy of community notes was, on average, 9.6% larger for reposters who had no prior interactions with the authors of misleading posts (ATT: −0.682; 99% CI: [−0.703, −0.660]; $z = -43.70$, $p < 0.001$), compared to reposters who had prior interactions (ATT: −0.586; 99% CI: [−0.608, −0.561]; $z = -40.10$, $p < 0.001$). This suggests that community fact-checks may be less effective in tackling misinformation disseminated by influential accounts with dedicated and trusting audiences (e.g., politicians, public figures).

In contrast, we did not find statistically significant differences in sharing behavior between verified (ATT: −0.595; 99% CI: [−0.640, −0.545]; $z = -20.00$, $p < 0.001$) vs. non-verified reposters (ATT: −0.634; 99% CI: [−0.653, $z = -50.24$, −0.615]; $p < 0.001$), left-leaning (ATT: −0.625; 99% CI: [−0.654, −0.595]; $z = -32.19$, $p < 0.001$) vs. right-leaning reposters (ATT: −0.626; 99% CI: [−0.652, −0.598]; $z = -35.34$, $p < 0.001$), and reposters with high (ATT: −0.633; 99% CI: [−0.655, −0.609]; $z = -40.63$, $p < 0.001$) vs. low exposure to misinformation (ATT:

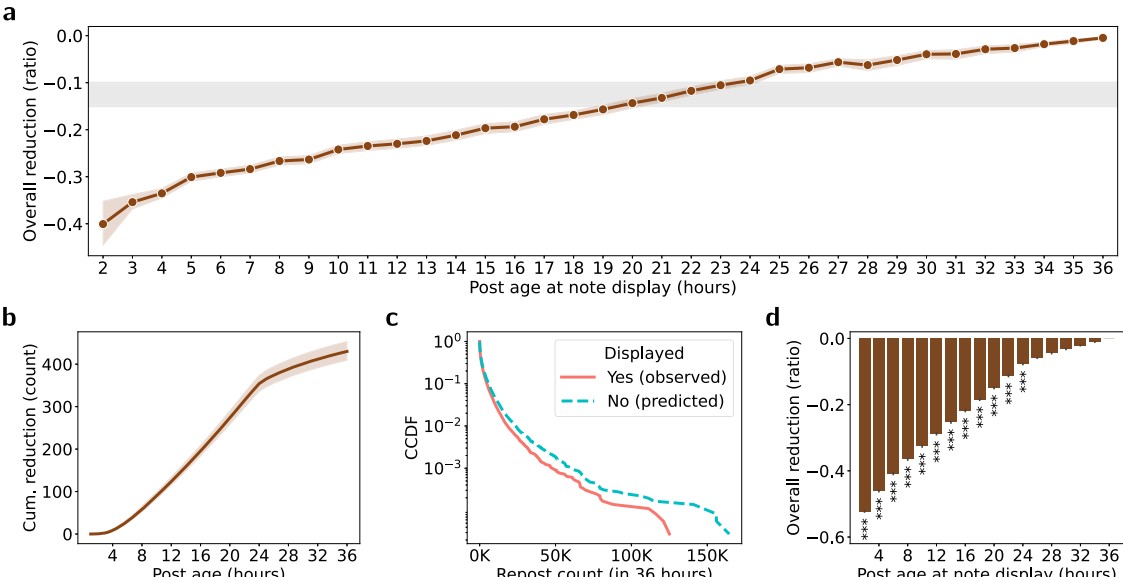

**Fig. 3 | Effect of community notes on the cumulative repost count of misleading posts on X (formerly Twitter). a** The changes in the ratio of the reduction of reposts relative to the predicted overall repost count over the response time from post creation to note display (circles; grouped within 1-h windows). The error band represents 99% CIs derived using the bootstrap method for 1000 resamples. The grey band ranges between the mean and the median of the ratio of the reduction. **b** The estimated cumulative count of reposts that community notes prevents at different post ages. The error band represents 99% CIs derived using the bootstrap method for 1000 resamples. **c** CCDFs showing the actually observed repost count for source posts with displayed community notes and the predicted repost count that the source posts would have received in the absence of community notes display. **d** The estimated overall reduction in reposts if all posts with community notes would have been displayed simultaneously from 2 to 36 h after post creation. Statistical significance (*$p < 0.01$; **$p < 0.005$; and ***$p < 0.001$) was calculated using two-tailed KS tests (see detailed results in Table S33 and Supplementary Note 8).

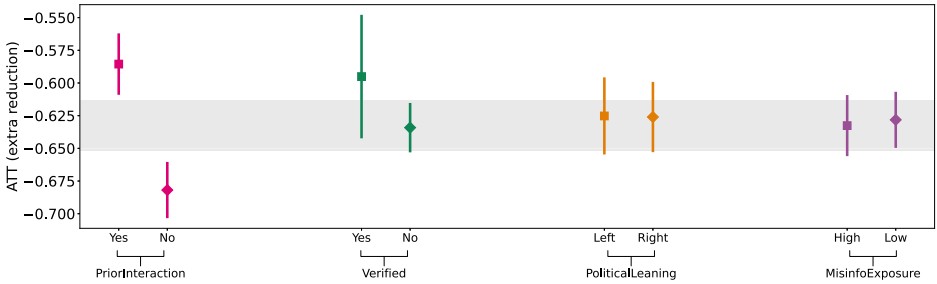

**Fig. 4 | Analysis of reposting mechanisms.** The estimated ATTs (circles) across different groups of reposters. Here we distinguish between (i) reposters that had prior interactions with the authors of the fact-checked post, (ii) the verification status of reposters, (iii) the political orientation of reposters, and (iv) prior exposure of reposters to misinformation on X (formerly Twitter). The ATT estimations are based on a random subset of 50,608 repost time series observations for $N = 3136$ posts. Post-level random effects are included. Full estimation results are in Supplementary Note 9. In all plots, the error bars represent 99% CIs derived using the bootstrap method for 1000 resamples; the grey bands visualize the ATT (with 99% CIs) estimated via the two-period DiD model based on reposts from all categories of reposters.

−0.628; 99% CI: [−0.649, −0.606]; $z = −44.24$, $p < 0.001$). This implies that displaying community fact-checks on misleading posts is broadly effective across both sides of the political spectrum and that even people who are steeped in misinformation are responsive to community fact-checks.

### Effect on post deletion and suspension/protection

We further examined the effect of community notes on the likelihood of deletion of misleading posts by their authors. Notably, the X (formerly Twitter) API only provided us with the deletion status and did not offer access to information regarding the time of deletion. Therefore, we did not know whether the posts were deleted before or after the display of community notes. Given this, we employed regression discontinuity design (RDD) and hypothesized that, in the absence of a significant effect of community notes, the probability of post-deletion should have exhibited smooth continuity (or keep

stable) around the threshold of the note helpfulness score that determined the display of community notes (see Supplementary Note 11 for details). As of the date of data collection, the cut-off point (threshold) for the note helpfulness score was 0.40, meaning that only community notes receiving a score of 0.40 or above are eligible to be displayed on the corresponding misleading posts. Thus, in our RDD model, observations are assigned to the treatment or control group based on their values of the note helpfulness score, and the treatment effect is estimated by comparing the outcomes just above and just below the cutoff point.

The estimated changes in the ratio of deleted posts for different note helpfulness scores are visualized in Fig. 5a. We observed a clear, sharp discontinuity at the cut-off point of 0.40. Specifically, the estimated treatment effect of community notes display in our RDD model was 0.943 (99% CI: [0.611, 1.342]; $z = 9.15$, $p < 0.001$), which implies that the odds for deletion by their authors were 94.3% higher for posts with

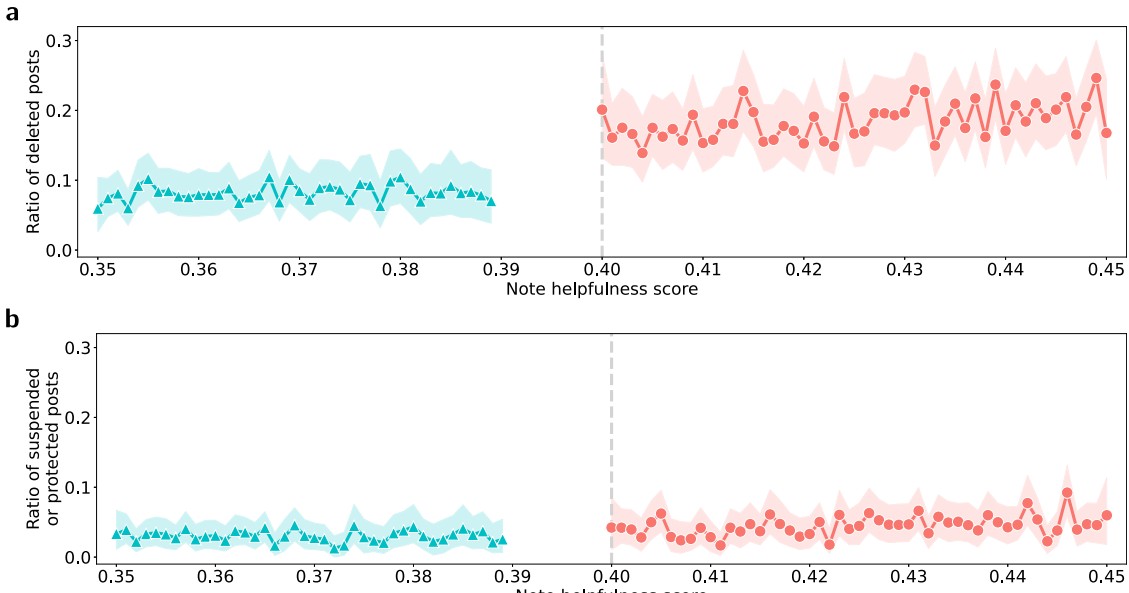

**Fig. 5 | Effect of community notes on post deletion and suspension/protection. a** Shown are the ratios of deleted posts for different values of note helpfulness (circles). **b** Shown are the ratios of protected/suspended posts for different values of note helpfulness (circles). Note helpfulness scores are centered around the cutoff point of 0.40. Only community notes with note helpfulness scores of 0.40 or above are displayed on the corresponding misleading posts. The note helpfulness scores are rounded to three decimal places. Notes with helpfulness scores of 0.39 are omitted to prevent treatment contamination due to fluctuations between recalculated note helpfulness scores and note helpfulness scores used in production. The error bands represent 99% CIs derived using the bootstrap method for 1000 resamples. The estimations are based on $N = 36{,}136$ posts. See Supplementary Note 11 for further details and full estimation results.

displayed community notes than for posts without displayed community notes (see Supplementary Note 11).

In addition to deletion, misleading posts with displayed notes could become inaccessible either because they were protected by their authors (i.e., private posts) or due to account suspensions imposed by the X (formerly Twitter) platform. Although the X (formerly Twitter) API does not distinguish between these two scenarios, a manual review of 1500 posts revealed that almost all (94.4%) of these inaccessible posts resulted from account suspensions (i.e., platform-enforced actions). We adopted the same RDD approach to examine the potential effect of community notes on post- suspension/protection (see Supplementary Note 11 for details). Unlike the findings for post deletions, Fig. 5b shows that there was no sharp change in the ratio of suspended/protected posts around the cut-off point of note helpfulness scores. Furthermore, the coefficient estimate for the treatment effect of community notes on post-suspension/protection was not statistically significant (see Supplementary Note 11), reinforcing this observation. In summary, our analysis provides confirmatory evidence that community notes did not trigger enforcement of other content moderation rules on X (formerly Twitter), i.e., platform-enforced actions (see Supplementary Note 12 for details).

## Discussion

We performed a large-scale quasi-experimental study to assess the real-world efficacy of community-based fact-checking on the social media platform X (formerly Twitter). Based on time series data for $N = 237{,}180$ (community fact-checked) X (formerly Twitter) cascades that had been reposted more than 431 million times, we found that exposing users to community notes reduced the spread of misleading posts by, on average, 61.2%. This is largely consistent with small-scale A/B tests carried out by X (formerly Twitter) internally during the pilot phase of community notes[66]. Furthermore, the odds for post deletion were 94.3% higher for posts with displayed community notes than for posts without displayed community notes. Crucially, X (formerly Twitter)'s recommendation algorithm (available as open source[67]) does not impose any penalties (e.g., visibility reduction) on posts

flagged with community notes[68]. This implies that the observed effects must be attributed to shifts in user behavior, rather than algorithmic intervention (Supplementary Note 12). Altogether, our findings complement earlier studies focusing on the question of whether crowds are able to accurately assess social media content[33,37–43], by offering real-world empirical evidence that community notes reduces the sharing of misleading posts on X (formerly Twitter).

The treatment effect of community notes was pronounced and statistically significant across the board. Individuals across both sides of the political spectrum, even those who are frequent consumers of misinformation, were less likely to share misleading posts when informed by community notes. Notwithstanding, we still observed substantial variations concerning both user characteristics and content characteristics. Specifically, the efficacy of community-based fact-checking was discounted for posts originating from influential accounts (e.g., verified users, high-follower accounts). A potential explanation is that such users may have the ability to attract more like-minded individuals, fostering echo chambers where their false beliefs are reinforced[69], thereby potentially mitigating the efficacy of community notes. Consistent with this notion, our results indicate that a primary mechanism through which community fact-checks reduce engagement is by discouraging resharing among users who had no prior interaction with the author of the misleading posts (i.e., disconnected users). Put differently, this implies that community fact-checks may be less effective in tackling misinformation disseminated by influential accounts with dedicated and trusting audiences (e.g., politicians, public figures). In addition to user heterogeneity, content characteristics also significantly influenced the efficacy of community notes. The presence of attached media items (e.g., images, videos) significantly amplified the intervention effect. In contrast, community notes were least efficacious when addressing health-related or political misinformation.

While community notes exhibited a pronounced treatment effect, the timing of their display poses a significant challenge. Community-based fact-checking tends to be faster than professional fact-checking[51], yet the rapid dissemination of posts on X (formerly Twitter) still outpaced the (relatively) slow display of community notes,

limiting the feature's ability to curb misinformation early on. In our data, the half-life of reposts over 36 h is 5.75 h, with only 13.5% of all helpful notes displayed before this time point. Such delays reduced their effectiveness during the early, and often most viral, stages of diffusion. As a result, their system-wide effect was more modest, lowering the total number of reposts of misleading posts by 14.9%. This effect was smaller during the early phases of the community notes program, consistent with ref. 57 who analyze aggregated engagement metrics from before and shortly after the feature's rollout (see Supplementary Note 10.4), but has grown stronger over time. Notably, the observed effect sizes are also broadly in line with earlier evidence on the efficacy of fact-checking interventions. Internal data from meta suggested that labels on false election posts reduced sharing by ≈8% cumulatively, whereas the company's self-reported labeling and downranking of fact-checked false content was claimed to reduce subsequent sharing by ≈80%[70]. By comparison, community notes achieved a 61.2% decrease in subsequent sharing, even without downranking measures. Altogether, the key implication is thus that community notes are effective in reducing the spread of posts once annotated, but their system-wide impact depends critically on speed: notes must appear early, ideally before misleading posts gain traction and spread widely.

As with any study, ours is not without limitations, which present opportunities for future research. First, our focus on community-based fact-checking was limited to one social media platform, specifically examining the effectiveness of community notes on X (formerly Twitter). While this platform represents the only large-scale implementation of community-based fact-checking on mainstream social media platforms to date, future research could expand to include other platforms if they choose to implement similar features (e.g., YouTube and Meta have recently begun testing a fact-checking feature comparable to X (formerly Twitter)'s Community Notes). Second, the community notes feature itself may evolve over time, potentially influencing its effectiveness in combating misinformation. While we took intensive care to account for potential effects of algorithmic changes in our empirical analysis (see Supplementary Note 12), future research should examine how further developments or enhancements to this feature influence its efficacy. Third, incorporating information on network structure could yield additional insights. For example, recent research[71] confirms our findings and shows that note attachment qualitatively changes the structure of the post's diffusion cascade. Fourth, our study primarily focused on the American context and the global North, with a specific emphasis on posts in English. Future research could explore the efficacy of community-based fact-checking across different cultural contexts to gain a more comprehensive understanding of its impact on diverse communities. Finally, it could be valuable to investigate how the intervention effect varies across different demographic groups, such as age, gender, or education level, to uncover insights into who is most influenced by community-based fact-checking efforts.

Fact-checking becomes increasingly vital, particularly in light of emerging challenges posed by the scalability of AI-generated misinformation. Countermeasures against misinformation are more relevant than ever, and community-based interventions, such as community notes, hold promise in this fight, at least in concept. However, the effectiveness of such interventions hinges on their ability to respond swiftly to the dynamic nature of misinformation dissemination. Therefore, enhancing the speed and efficacy of community-based fact-checking approaches is crucial in fortifying our defenses against the spread of false information in an era marked by heightened digital influence.

## Methods

Our study has received ethical approval from the Ethics Review Panel of the University of Luxembourg (ref. ERP 23-053 REMEDIS), and all data collection from X (formerly Twitter) complied with the platform's developer rules.

### Data collection

We downloaded all community note contributions via a dedicated website on July 14, 2024[72]. X (formerly Twitter) routinely updates the community note contributions on a daily base and releases them to the public via this website. The community note contributions were documented in three distinct datasets: notes, ratings, and note status history. Each note in the notes dataset includes its unique note ID, referenced post ID, creation timestamp, and classification (i.e., "misinformed or potentially misleading" or "not misleading"). The ratings dataset includes all the ratings submitted by the contributors along with their creation timestamps. The note status history dataset records the changes in the status of referenced notes and corresponding timestamps. The recorded note statuses contain the information on the helpfulness ratings, i.e., whether the notes were displayed on the corresponding misleading posts.

Based on the referenced post IDs in the notes dataset, we collected the corresponding source posts via the X (formerly Twitter) API v2 post lookup endpoint. Here, we only considered source posts written in English. Each post object returned from this endpoint contains detailed information related to the post itself (such as text, creation time, and media metadata) and its author profile (such as verified status, number of followers, and number of followees). Subsequently, we employed the X (formerly Twitter) API v2 full-archive post counts endpoint to collect time series data of repost counts at the minute level for the community fact-checked posts over a period of 36 hours since their creation. In total, our dataset contains time series data of repost counts for a total of $N = 237,180$ (community fact-checked) posts that were created between October 6, 2022 and June 11, 2024 and had been reshared more than 431 million times on X (formerly Twitter). A detailed overview of the dataset and descriptive statistics are available in Supplementary Notes 1 and 2, respectively.

### Empirical analysis

We employed a difference-in-differences design to estimate the treatment effect of community notes on the spread of misleading posts on X (formerly Twitter) as measured by the repost count over time following the note display. The treatment group consists of all source posts with displayed notes, i.e., posts that received a community note rated as helpful and, thus, displayed to all users on X (formerly Twitter). We considered the time of the display of the first helpful note for each post as the start of the treatment.

To ensure comparability, we constructed a control group from the source posts that share similar characteristics with the source posts in the treatment group but have no displayed notes. Given that the number of source posts without displayed notes (201,098) is nearly 6 times the number of source posts with displayed notes (36,082), we performed one-to-one propensity score matching for source posts with displayed notes and find their closest matches in the source post without displayed notes to construct a balanced control group (see Supplementary Note 4). The one-to-one matching was conducted based on the variables from user profiles (e.g., followers and followees) and post features (e.g., sentiments and topics). As a result, the source posts in the control group have no statistically significant differences from the source posts in the treatment group with respect to the user and post characteristics. Subsequently, we assigned the virtual time of note display to the source posts in the control group according to the corresponding posts in the treatment group and recentered the repost timelines around the display of community notes.

**Two-period DiD**. We implemented a two-period DiD model to estimate differences in repost counts between the treatment and control group

before and after the display of community notes. In our data, community notes were displayed 1.239 h after post creation at the earliest, and only 5.2% of community notes were displayed within 4 h since post creation. Therefore, we considered a 4-h window before the note display as the before-display period. The after-display period was set to between 1 and 12 h from the note display. The dependent variable in the DiD model is the number of reposts (i.e., RepostCount). Given that RepostCount is a nonnegative count variable with overdispersion, we employed a multilevel mixed-effects negative binomial regression. Formally, we specified the negative binomial regression model with post-level random effects (i.e., random intercepts) for the two-period ATT estimation as:

$$
\begin{aligned}
\log\left(\mathbb{E}[\text{RepostCount}_{it}|\mathbf{x}_{it}]\right) = {} & \beta_0 + \beta_1\,\text{Display}_i + \beta_2\,\text{After}_t \\
& + \beta_3\,(\text{Display}_i \times \text{After}_t) \\
& + \beta_4\,\text{PostAge}_{it} + \mu_{\text{post}} ,
\end{aligned} \tag{1}
$$

where $\mathbf{x}_{it}$ indicates all the independent variables and $\beta_0$ is the intercept. $\text{Display}_i$ is a group-specific dummy variable indicating whether the source post $i$ belongs to the treatment group (=1) or control group (=0). $\text{After}_t$ is a time-specific dummy variable indicating the time is before (=0) or after the note display (=1). $\text{Display}_i \times \text{After}_t$ is the difference-in-differences interaction capturing the treatment effect. Further, $\text{PostAge}_{it}$ denotes the age of source post $i$ at time $t$ since its creation. The variable $\mu_{post}$ represents post-specific random effects, which capture post-level heterogeneity that cannot be reflected by the general post features in the propensity score matching. For example, novel posts related to real-world events that happened suddenly may gain more engagement.

In our main analysis, we opted to use post-level random effects rather than fixed effects for the following reasons: (i) we conducted one-to-one propensity score matching to construct a control group that was balanced with the treatment group across all poster and post characteristics (see Supplementary Note 4). This effectively mitigates concerns that unobserved poster-specific or post-specific factors could be related to covariates. (ii) Our estimation was based on a large scale of 654,400 longitudinal observations across $N = 40,900$ posts, where the assumption of normal distribution of post-level random effects was appropriate. (iii) Fixed-effects models would omit posts with constant outcomes, i.e., repost counts, over time. Given that the distribution of repost counts were overdispersed with many observations at zero, dropping these constant observations would compromise the representativeness of our dataset. Nonetheless, we also repeated our analysis using post-level fixed effects, where we observed consistent results (see Supplementary Note 10.1). Additionally, to account for the possibility that excess zeros in repost counts at later stages of diffusion, we repeated our analysis with zero-inflated negative binomial regression models. Also here, the results were robust and consistent with our main analysis (see Supplementary Note 10.2).

**Multi-period DiD**. We estimated a multi-period DiD to enable a temporal analysis of the treatment effect. Here, we again considered a 4-h window before the note display as the before-display period and then examined the hourly multi-period ATTs from 1 to 12 h after the note display. The baseline of time effects is the first hour before note display. Formally, we specified the negative binomial regression model for the multi-period ATT estimation as:

$$
\begin{aligned}
\log\left(\mathbb{E}[\text{RepostCount}_{it}|\mathbf{x}_{it}]\right) = {} & \beta_0 + \beta_1\,\text{Display}_i + \mathbf{b}_1^\top\,\mathbf{Before}_t + \mathbf{b}_2^\top\,\mathbf{After}_t \\
& + \mathbf{b}_3^\top\,(\text{Display}_i \times \mathbf{Before}_t) \\
& + \mathbf{b}_4^\top\,(\text{Display}_i \times \mathbf{After}_t) \\
& + \beta_2\,\text{PostAge}_{it} + \mu_{\text{post}} ,
\end{aligned} \tag{2}
$$

where the vector $\mathbf{b}_3$ includes the coefficient estimates for the parallel test during the before-display period, and $\mathbf{b}_4$ represents the coefficient estimates for the corresponding difference-in-difference interactions that can be transformed to hourly multi-period ATTs. All other variables are the same as those in the two-period model.

**Calculation of ATTs**. The coefficients estimated for the difference-in-differences terms in the two-period and multi-period regression models are the natural logarithms of the ratios for the number of reposts with the treatment of note display compared to the number of reposts that are expected to receive without the note display during the after-display period. We exponentially transformed the coefficient estimates of the difference-in-differences terms in the models and measured the treatment effect of community notes (i.e., ATT) as:

$$
\text{ATT} = e^{\beta} - 1, \tag{3}
$$

where $\beta$ is the coefficient estimate for the specific difference-in-differences term. ATT indicates the ratio of extra change of the reposts in the source posts with displayed notes relative to the reposts that the source posts are expected to receive without displayed notes. We used this indicator to analyze the efficacy of community notes.

**Implementation**. We used Python 3.11.3 to conduct our empirical analyses. Our regression models were implemented using the pystata Python package with StataNow 19.5 MP-parallel edition (2-core network).

### Ethics statement
This research has received ethical approval from the Ethics Review Panel of the University of Luxembourg (ref. ERP 23-053 REMEDIS). All analyses are based on publicly available data.

### Reporting summary
Further information on research design is available in the Nature Portfolio Reporting Summary linked to this article.

## Data availability
The X (formerly Twitter) data collected and analyzed in this study have been deposited in an Open Science Framework (OSF) repository (https://doi.org/10.17605/OSF.IO/M642D). Due to data protection considerations and compliance with X (formerly Twitter)'s API usage agreement, we cannot disclose individual-level user information or the contents of posts. Instead, processed and pseudonymized data are available in the repository to support the replication of the analyses and results reported in this study.

## Code availability
The code to replicate the findings of our study is available at https://doi.org/10.17605/OSF.IO/M642D.

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

## Acknowledgements

This study was supported by research grants from the German Research Foundation (N.P.; DFG grant 492310022), the Luxembourg National Research Fund (G.L.; INTER_FNRS_21_16554939_REMEDIS), the HEC Paris Foundation (T.R.), and the Hi! Paris Center for Data Analytics and Artificial Intelligence (T.R.). The funders had no role in study design, data collection and analysis, decision to publish, or preparation of the manuscript.

## Author contributions

Y.C., M.P., T.R., D.R., A.T., G.L., and N.P. conceived and designed the experiments. Y.C., M.P., and T.R. collected the data. Y.C., M.P., and T.R. analyzed the data. Y.C., M.P., T.R., D.R., A.T., G.L., and N.P. wrote the manuscript. All authors approved the manuscript.

## Funding

## Competing interests
The authors declare no competing interests.

## Additional information
**Supplementary information** The online version contains Supplementary material available at https://doi.org/10.1038/s41467-026-72597-0.

