## [Transparent Peer Review File · Nature Communications]

Community-based fact-checking reduces the spread of misleading posts on X (formerly Twitter)

Corresponding Author: Professor Nicolas Pröllochs

Version 0:

Reviewer comments:

Reviewer #2

(Remarks to the Author)

The authors have satisfactorily addressed my comments from the prior round.

Reviewer #3

(Remarks to the Author)

I served as the reviewer on a previous version of this paper. I am happy to see that the authors have incorporated all of the changes laid out in my earlier review, including

- Using Robust DiD methods
- Sensitivity testing of the 4 hour cutoff
- Engaging with the authors' previous work
- Streamlining the results and moving elements to supplement

The paper seems much improved based on these edits.

However, I have some outstanding questions:

- Is it possible to quantify the cumulative effect with robust HonestDiD methods?
- Are the reposts in the "Analysis of Reposting Mechanisms" analysis from after the community notes are shown, or throughout the entire lifespan of the post? Is it possible that the "prior interaction" subset are just seeing the tweets before the note was appended? Either way it's interesting, but the analysis was not clear.
- Substantively, the paper to me seems a little bit overly critical of the Community Notes program, e.g. saying the cumulative effects are "merely" 15%. I think the authors could do more to contextualize these effects, which are in line – and perhaps better – than what other interventions have found. While the points about speed are very important, I think it's relevant that work has found that Community fact-checking is actually faster than professional fact-checking (Zhao & Naaman 2023) and provides more coverage. And while 15% isn't enormous, leaked information from Meta found that labels on Trump's false election posts reduced sharing by only ~8% cumulatively (Silverman 2020). While Meta's labeling and downranking of fact-checked false content lowered sharing by 80% (according to them), this labeling often occurred after the post had already spread and is not that different from the 60% decrease that the authors find without any downranking measures.

Minor points:

- I'm not sure if the researchers have seen the recent Slaughter et al (2025) preprint that tackles a similar analysis of Community Notes, but they might want to cite it. It is encouraging that the two have similar findings

<https://arxiv.org/pdf/2502.13322>

- Small point: when the authors say "the ATT estimates were higher" on page 13, it implied (to me) that the magnitude of the effects were lower (since the coefficients are negative), even though the opposite is true. Suggest that the authors instead say "larger"

[1] Zhao, A., & Naaman, M. (2023). Insights from a comparative study on the variety, velocity, veracity, and viability of crowdsourced and professional fact-checking services. *Journal of Online Trust and Safety*, 2(1).

[2] Slaughter, I., Peytavin, A., Ugander, J., & Saveski, M. (2025). Community Notes Moderate Engagement With and Diffusion of False Information Online. arXiv preprint arXiv:2502.13322.

[3] Silverman, Craig, and Ryan Mac. "Facebook Knows That Adding Labels to Trump's False Claims Does Little to Stop Their Spread." *BuzzFeed News* 16 (2020). <https://www.buzzfeednews.com/article/craigsilverman/facebook-labels-trump-lies-do-not-stop-spread>

Comments on response to Reviewer 1's previous concerns:

I think that the proposed analysis by the authors of analyzing around the cutout does hypothetically address the issue, but I am a bit confused by the setup, since it's not a traditional RD design with a running variable despite the authors' writing that it is a "Regression Discontinuity (RD)-based restriction." Can the authors do a traditional RD, like they did in section J.1? And include the typical RD plot of the running variable (note helpfulness score) on the X-axis, with total engagement on the y-axis, and the before vs after periods marked? That will better allow for understanding.

One reason why I'm concerned with the current robustness check is that the display coefficient is significant and negative, suggesting that the not-helpful notes are different from the helpful notes in the pre-period, giving credence to R1's concern that the two groups are different in ways not only attributable to the display of a note. If the authors could add a plot like Figure 1 that would be nice.

Version 1:

Reviewer comments:

Reviewer #3

(Remarks to the Author)

I am satisfied with the authors' edits and would support acceptance of this paper.

Revised submission to Nature Communications (NCOMMS-25-21301-T)

Manuscript: “*Community-based fact-checking reduces the spread of misleading posts on social media*”

Dear editors, dear reviewers,

Thank you very much for considering our paper for possible publication at *Nature Communications*. On the road to finalizing our paper, we are deeply grateful for the positive remarks and the additional feedback regarding specific areas in need of improvement.

We have followed all comments and suggestions closely to remedy the remaining points raised by the reviewers. In the following, we will address all points individually and show how each issue is resolved. To facilitate the review, changes in the revised paper are highlighted in **blue** text color.

We appreciate your time and look forward to hearing your feedback. If you have any questions, please do not hesitate to contact us.

Kind regards,

Yuwei Chuai, Moritz Pilarski, Thomas Renault, David Restrepo-Amariles, Aurore Troussel-Clément, Gabriele Lenzini, and Nicolas Pröllochs

RESPONSES TO REVIEWER #1

Comment R1.1: *The authors have satisfactorily addressed my comments from the prior round.*

Response R1.1: Thank you very much for the positive feedback. We appreciate your detailed comments and suggestions throughout the review process.

RESPONSES TO REVIEWER #2

Comment R2.1: *I served as the reviewer on a previous version of this paper. I am happy to see that the authors have incorporated all of the changes laid out in my earlier review, including*

- *Using Robust DiD methods*
- *Sensitivity testing of the 4 hour cutoff*
- *Engaging with the authors' previous work*
- *Streamlining the results and moving elements to supplement*

The paper seems much improved based on these edits.

Response R2.1: We thank the reviewer for their careful reading of our revised manuscript and for recognizing the changes we made in response to the earlier review. We are grateful for the constructive feedback throughout the review process, which significantly improved our paper.

Comment R2.2: *However, I have some outstanding questions:*

- *Is it possible to quantify the cumulative effect with robust HonestDiD methods?*

Response R2.2: We thank the reviewer for this suggestion. In the revised manuscript (see SI, Sec. I.6), we now quantify the cumulative effect using robust HonestDiD methods. We find that the results are highly consistent with our main estimates (cumulative effect of 14.4% vs. 14.9%).

Comment R2.3: - *Are the reposts in the "Analysis of Reposting Mechanisms" analysis from after the community notes are shown, or throughout the entire lifespan of the post? Is it possible that the "prior interaction" subset are just seeing the tweets before the note was appended? Either way it's interesting, but the analysis was not clear.*

Response R2.3: The prior interactions metric captures the total number of interactions (reposts) a user had with the original poster during the 28 days preceding the publication of the tweet targeted by a Community Note. It does not include interactions with the tweet itself prior to the addition of a note. For our analysis of reposting mechanisms, we collected reposts within a fixed window of -4 to +12 hours relative to the post's publication (i.e., time series data), which means that both reposts before and after the display of a community note are included but those are not used to construct the "prior interaction variable". To address the reviewer's concern, our ATT estimations are explicitly based on the after-display period, while the pre-display reposts are used only to establish parallel pre-trends. Thus, the "prior interaction" subset does not merely capture users who reposted before the note was appended; rather, the treatment effects we report reflect differences in reposting behavior after notes were displayed, conditional on prior interaction. We have clarified this in the revised manuscript (see SI, Section H).

Comment R2.4: - *Substantively, the paper to me seems a little bit overly critical of the Community Notes program, e.g. saying the cumulative effects are “merely” 15%. I think the authors could do more to contextualize these effects, which are in line – and perhaps better – than what other interventions have found. While the points about speed are very important, I think it’s relevant that work has found that Community fact-checking is actually faster than professional fact-checking (Zhao & Naaman 2023) and provides more coverage. And while 15% isn’t enormous, leaked information from Meta found that labels on Trump’s false election posts reduced sharing by only ~8% cumulatively (Silverman 2020). While Meta’s labeling and downranking of fact-checked false content lowered sharing by 80% (according to them), this labeling often occurred after the post had already spread and is not that different from the 60% decrease that the authors find without any downranking measures.*

Response R2.4: We appreciate this thoughtful comment and agree that parts of the original text were overly critical of the Community Notes program. In the revised discussion, we adjusted the language accordingly. Moreover, we added further context by comparing our findings to prior work, including evidence on the relative speed of community-based fact-checking, as well as a comparison to Meta’s fact-checking program (see revised Discussion).

Comment R2.5: *Minor points:*

- *I’m not sure if the researchers have seen the recent Slaughter et al (2025) preprint that tackles a similar analysis of Community Notes, but they might want to cite it. It is encouraging that the two have similar findings <https://arxiv.org/pdf/2502.13322>*

Response R2.5: We agree that it is encouraging that Slaughter et al. (2025) also find that Community Notes reduces engagement with false content, leading to a 46.1% drop in reposts, which is very close to the estimate reported in our paper. We now cite their work in the revised discussion and highlight the consistency between their findings and ours.

Comment R2.6: - *Small point: when the authors say “the ATT estimates were higher” on page 13, it implied (to me) that the magnitude of the effects were lower (since the coefficients are negative), even though the opposite is true. Suggest that the authors instead say “larger”*

Response R2.6: Thanks for spotting this. We now write “larger” instead of “higher,” as suggested.

Comment R2.7: *Comments on response to Reviewer 1’s previous concerns:*

I think that the proposed analysis by the authors of analyzing around the cutout does hypothetically address the issue, but I am a bit confused by the setup, since it’s not a traditional RD design with a running variable despite the authors’ writing that it is a “Regression Discontinuity (RD)-based restriction.” Can the authors do a traditional RD, like they did in section J.1? And include the typical RD plot of the running variable (note helpfulness score) on the X-axis, with total engagement on the y-axis, and the before vs after periods marked? That will better allow for understanding.

One reason why I'm concerned with the current robustness check is that the display coefficient is significant and negative, suggesting that the not-helpful notes are different from the helpful notes in the pre-period, giving credence to R1's concern that the two groups are different in ways not only attributable to the display of a note. If the authors could add a plot like Figure 1 that would be nice.

Response R2.7: We thank the reviewer for these thoughtful suggestions. As acknowledged in our previous response to R1, we fully concur that, in the absence of an RCT, posts receiving and not receiving Community Notes may differ in ways that we cannot fully control for.

Notwithstanding, based on your feedback, we have taken additional steps to further address this concern and strengthen the robustness of our analysis. Specifically, we have conducted additional robustness checks restricting the sample based on both note helpfulness scores and pre-display engagement. Specifically, we (1) limited the analysis to posts with note helpfulness scores within a narrow window around the display threshold (0.35–0.45, cutoff at 0.40) to mitigate potential confounding related to note scoring. (2) To ensure comparability in engagement prior to note display, we matched each control post with a treatment post whose pre-display repost activity differed by less than 0.1%.

Within this restricted sample, we assessed the observed diffusion of treated posts in comparison to the counterfactual diffusion of their matched counterparts. Specifically, we estimated the treatment effect using two complementary approaches: a Difference-in-Differences (DiD) design and a Regression Discontinuity Design (RDD) based on note helpfulness scores.

- **Difference-in-Differences (DiD).** We re-estimated the treatment effect for the restricted sample using a DiD design analogous to our main specification (see coefficient estimates in Fig. S14a). With the restricted sample, the display coefficient was no longer statistically significant. The two-period ATT estimate was -0.595 (99% CI: $[-0.617, -0.572]$; $p < 0.001$), which is nearly identical to the results from our main analysis, reinforcing the robustness of our findings.
- **Regression Discontinuity Design (RDD).** We further implemented a traditional RDD using note helpfulness scores as the running variable, as suggested. As shown in Fig. S14b, the data shows a clear drop in after-display reposts (the total number of reposts during the 12-hour after-display period) after the cutoff point of 0.40. The RDD estimate (-0.616 , 99% CI: $[-0.640, -0.591]$) closely matches the DiD estimate, providing strong confirmatory evidence for the robustness of our results.

In summary, these findings provide strong confirmatory evidence that the observed reduction in engagement is attributable to the presence of Community Notes rather than to systematic differences between noted and non-noted posts.

The new analyses and results are in the SI, Section I.3.